# VIDEO-MTR: REINFORCED MULTI-TURN REASONING FOR LONG VIDEO UNDERSTANDING

## ABSTRACT

Long-form video understanding, characterized by long-range temporal dependencies and multiple events, remains a challenge. Existing methods often rely on static reasoning or external visual-language models (VLMs), which face issues like complexity and sub-optimal performance due to the lack of end-to-end training. In this paper, we propose Video-MTR, a reinforced multi-turn reasoning framework designed to enable iterative key video segment selection and question comprehension. Unlike traditional video reasoning pipeline, which generates predictions in a single turn, Video-MTR performs reasoning in multiple turns, selecting video segments progressively based on the evolving understanding of previously processed segments and the current question. This iterative process allows for a more refined and contextually aware analysis of the video. To ensure intermediate reasoning process, we introduce a novel gated bi-level reward system, combining trajectory-level rewards based on answer correctness and turn-level rewards emphasizing frame-query relevance. This system optimizes both video segment selection and question comprehension, eliminating the need for external VLMs and allowing end-to-end training. Extensive experiments on benchmarks like VideoMME, MLVU, LongVideoBench, LVBench and EgoSchema demonstrate that Video-MTR outperforms existing methods in both accuracy and efficiency, advancing the state-of-the-art in long video understanding.

## 1 INTRODUCTION

As a foundational computer vision task, video understanding finds widespread applications in numerous domains ranging from intelligent surveillance, content-based retrieval, to autonomous driving. With the explosive growth of user-generated videos and the ubiquity of cameras in daily life, the demand for robust and scalable video-understanding tools has grown substantially. Owing to the advanced reasoning capabilities, Multimodal Large Language Models (MLLMs) (Dai et al., 2023; Wu & Xie, 2024; Weng et al., 2024; Chen et al., 2024b) have demonstrated breakthroughs in visual understanding tasks for images and short videos in recent years. However, long-form video understanding(LVU), characterized by multiple events and long-range temporal dependencies, still presents significant challenges.

Existing approaches (Wang et al., 2024c; Lin et al., 2023; Feng et al., 2025) either employ instruction tuning or integrate reinforcement learning to adapt current MLLMs for long-term temporal reasoning. However, these methods primarily transfer training paradigms designed for language and image modalities, relying on a static reasoning approach that generates predictions based on a fixed, uniform set of sampled frames in a single turn. This single-turn, uniform sampling strategy creates a bottleneck for downstream reasoning tasks when dealing with long-form videos, as it risks omitting critical information due to the extended video duration. Alternatively, other approaches (Fan et al., 2024; Wang et al., 2024b; Ma et al., 2025) explore the agentic paradigm, where large language models (LLMs) serve as agents, utilizing external visual-language models (VLMs) (Radford et al., 2021; Zhao et al., 2023) to identify key video segments. These methods depend on pretrained VLMs and carefully designed pipelines. While they achieve superior performance, they are hindered by high complexity due to the reliance on heterogeneous external components and sub-optimal tool usage strategies, as they lack end-to-end training.

In this work, we propose Video-MTR, a reinforced multi-turn reasoning framework that leverages the intrinsic capabilities of MLLMs, equipped with bi-level rewards, for iterative key video segment selection and question comprehension within a unified model. Unlike existing video reasoning models, Video-MTR enables iterative selection of key video segments based on the current state, derived from previously selected segments and the question. This approach facilitates the progressive identification of more informative video segments. Compared to the agentic paradigm, Video-MTR eliminates the reliance on external VLMs and carefully designed pipelines, enabling end-to-end training that optimizes video segment selection and, in turn, enhances question comprehension.

Formally, Video-MTR builds upon an existing MLLM model, Qwen2.5-VL-7B (Bai et al., 2025) and is trained to develop iterative video reasoning capabilities through an end-to-end reinforcement learning strategy. However, current reward systems based solely on answer accuracy offer limited guidance for intermediate video segment selection, particularly in complex long videos. To address this challenge, we introduce a novel gated bi-level reward system, consisting of trajectory-level rewards based on answer correctness and turn-level rewards that capture frame-query relevance. This reward system relies on key segment annotations for turn-level rewards and the final answer for trajectory-level rewards. To enable this, we leverage the limited-scale QA-grounded corpus and augment it with a curated video temporal grounding dataset, using a tailored curation pipeline to align the original annotations with our QA-centric paradigm. Leveraging carefully designed reward functions, Video-MTR substantially alleviates reliance on large-scale datasets: whereas existing approaches typically require 256K-4.4M examples, Video-MTR achieves competitive or superior performance with only about 8K samples. Moreover, to maintain video understanding as the primary optimization objective, we anchor frame-level rewards exclusively to final answer correctness, enforcing that intermediate operations must genuinely contribute to the core task.

The contributions of this work are three-fold. First, we introduce Video-MTR, a reinforced multi-turn reasoning framework designed for long-form video understanding, enabling iterative video segment selection and question comprehension. To the best of our knowledge, this is the first attempt to incorporate end-to-end reinforcement learning with explicit multi-turn reasoning in this domain. Second, we propose a novel gated bi-level reward mechanism, which includes trajectory-level rewards based on answer correctness and turn-level rewards focused on frame-query relevance, facilitating more effective and informed video segment selection and substantially reduces dependence on large training corpora. Finally, we conduct extensive experiments on several video understanding benchmarks, including VideoMME (Fu et al., 2025), MLVU (Zhou et al., 2024), LongVideoBench(Wu et al., 2024), LVBench(Wang et al., 2024a) and EgoSchema (Mangalam et al., 2023), demonstrating the effectiveness and robustness of Video-MTR. Codes, trained models, and dataset will be released for further research.

## 2 RELATED WORK

### 2.1 MLLMS FOR VIDEO UNDERSTANDING

Building on image MLLMs' visual reasoning capabilities, researchers develop temporal extensions for video understanding. However, long-form videos remain challenging due to their extended duration exceeding contemporary MLLMs' context windows. Approaches like Video-LLaVA (Lin et al., 2023), ShareGPT4Video (Chen et al., 2024a), InternVideo2 (Wang et al., 2024c) and Video-R1(Feng et al., 2025) still resort to uniformly sampling the entire video and rely on post-training with large-scale video-instruction data to boost reasoning abilities. Yet the inevitable loss of information at the input stage creates a performance ceiling. Other approaches explicitly address this bottleneck. One category of methods, exemplified by LongVA (Zhang et al., 2024), LLaMA-VID (Li et al., 2024c), Kangaroo (Liu et al., 2024) and Video-XL (Shu et al., 2025), employs token compression techniques to extend context windows, enabling direct processing of hour-long videos. However, this approach floods the model with redundant information and sacrifices interpretability. Another category, like VideoAgent (Wang et al., 2024b), VideoMemAgent (Fan et al., 2024) and DrVideo (Ma et al., 2025) adopt agent mechanisms (Li et al., 2023; Wu et al., 2023) that dynamically integrate external tools, including video captioning, video object tracking, and key-frame search, through single-turn or multi-turn iterations. Despite outperforming uniform sampling baselines, these systems exhibit high complexity from heterogeneous external components and suboptimal tool utilization due to the absence of end-to-end training.

Figure 1: Overview of the proposed Video-MTR framework. *Left:* The lower part shows the multi-turn interaction loop between the MLLM agent and the video environment, while the upper part visualizes the collected trajectory and the gated bi-level reward shaping process during optimization. *Right:* Detailed logs of the agent's interaction steps across turns.

## 2.2 MLLMs with Reinforcement Learning

Recent studies (Shen et al., 2025; Meng et al., 2025), inspired by advances in the text domain, have explored reinforcement learning (RL) to improve the reasoning abilities of MLLMs. VLM-R1 (Shen et al., 2025) extends the DeepSeek-R1 paradigm (Guo et al., 2025), showing that an RL-trained MLLM can outperform a supervised fine-tuning baseline and generalize better on visual tasks. DeepEyes (Zheng et al., 2025) incentivizes "thinking with images" over multiple turns via RL. In the video domain, VideoChat-R1 (Li et al., 2025) enhances spatio-temporal perception through reinforcement fine-tuning (RFT) with GRPO, while Video-R1 (Feng et al., 2025) employs a tailored T-GRPO algorithm to emphasize temporal cues. However, these methods primarily target static images or short clips, leaving long-form video understanding largely unaddressed.

## 3 Methods

### 3.1 Overview

We propose Video-MTR, a framework that reconceptualizes long-form video understanding as a multi-turn interactive reasoning task, closely aligned with the way humans process complex visual information. When presented with a video and a question, humans typically begin by forming a holistic understanding of the overall content, then iteratively attend to specific segments to gather more informative details, and finally integrate the accumulated evidence to derive an answer.

To instantiate this reasoning paradigm, we formulate the task as a reinforcement learning problem. In this formulation, the video functions as a dynamic environment that updates the set of observed frames $\mathcal{F}$ in response to retrieval actions. An MLLM serves as the decision-making agent, interacting with the environment through a learned policy $\pi_\theta$. As illustrated in Figure 1, the agent operates in a multi-turn manner, and at each step it samples an action $a_k \sim \pi_\theta(\cdot|s_k)$ to either retrieve additional frames or produce the final answer. The state $s_k$ is a multimodal context that concatenates (i) the last $w$ interactions and (ii) the currently observed frames, providing both temporal history and updated visual evidence, and can be represented as

$$s_k = (\mathcal{F}_{k-w}, x_{k-w}, y_{k-w}, \ldots, \mathcal{F}_{k-1}, x_{k-1}, y_{k-1}, \mathcal{F}_k, x_k)$$

where $x$ is the text instruction, $\mathcal{F}$ is the set of observed frames, $y$ is the generated response that consists of reasoning process and executable action $a$. The environment is initialized by uniformly sampling $n_0$ frames to form $\mathcal{F}_0$ from the whole video. Thereafter, the environment responds to each retrieval action with a new set of frames that become the observation for the next turn. The agent may execute multiple retrieval actions until it is either confident enough to answer or the turn limit $K_{\max}$ is reached. The complete trajectory is recorded as:

$$\tau = \left\{ (\mathcal{F}_k, x_k, y_k) \right\}_{k=0}^K.$$

where $k$ indexes the turns starting from the initial turn $k = 0$, and $K$ denotes the terminal turn, with $0 \le K \le K_{\max}$.

The complete rollout process is outlined in Algorithm 1.

---

**Algorithm 1** Rollout of Multi-turn Reasoning Trajectory

---

**Input**: Long video $V$, Policy MLLM $\pi_\theta$, Question $x_0$, Input frame set $\mathcal{F}_0$, Maximum turn $K_{\max}$
**Output**: Final trajectory $\tau$
**Initialize:** $k \leftarrow 0$, rollout trajectory $\tau \leftarrow (\mathcal{F}_0, x_0)$

1: **while** $k < K_{\max}$ **do**
2:     Generate response $y_k \sim \pi_\theta(\cdot \mid s_k)$
3:     $\tau \leftarrow \tau + y_k$
4:     $\langle reason_k, a_k \rangle \leftarrow Parse(y_k)$
5:     **if** $a_k$ matches `"Retrieval"` format **then**
6:         Extract $(t_{start}, t_{end})$ from $a_k$
7:         $\mathcal{F}_{k+1} \leftarrow \text{RETRIEVEFRAMES}(V, t_{\text{start}}, t_{\text{end}})$
8:         $x_{k+1} \leftarrow x_0$                        ▷ question remains unchanged
9:         $\tau \leftarrow \tau + (\mathcal{F}_{k+1}, x_{k+1})$
10:    **else if** $a_k$ matches `"Answer"` format **then**
11:        **break**                              ▷ Get final answer
12:    **else**
13:        $x_k \leftarrow$ "Invalid action. Let me rethink."    ▷ Regenerate response for invalid action
14:        $\tau \leftarrow \tau + (x_k)$
15:    **end if**
16:    $k \leftarrow k + 1$
17: **end while**
18: Collect final trajectory $\tau$

---

While prior studies have applied reinforcement learning to MLLMs for temporal reasoning tasks, they predominantly adopt a single-turn reasoning settings. However, standard RL frameworks for MLLMs struggle with multi-turn optimization due to uniform credit assignment of sparse terminal rewards across turns. This hinders learning nuanced intermediate behaviors that are critical to final success. Furthermore, optimizing solely based on final-task accuracy generally demands extensive training data because terminal supervision is sparse. To address these multi-turn challenges, we introduce a gated bi-level reward system that augments conventional trajectory-level rewards with turn-level rewards. The turn-level rewards encode frame–query relevance, yielding more informative and discriminative signals. As most video question answering datasets provide only QA annotations, we increase data diversity by incorporating a video temporal grounding dataset and curating it to our QA-centric setup. Additionally, observing limited proactive frame retrieval in pretrained MLLMs, we adopt a dynamic exploration-bootstrapping strategy to encourage multi-turn evidence seeking.

### 3.2 GATED BI-LEVEL REWARD

This section details our fine-grained reward design for RL training. We first describe the computation of the basic bi-level reward. We then present a goal-gated mechanism that prioritizes trajectory-level signals over turn-level ones to align intermediate decisions with the final goal, fostering coherent, goal-oriented multi-turn reasoning.

#### 3.2.1 BI-LEVEL REWARD

This bi-level architecture comprises two complementary components: a trajectory-level reward $R_{acc}$ providing global supervision, and intermediate turn-level rewards to deliver localized feedback within individual turns. The trajectory-level reward $R_{acc}$ is binary, set to 1 if the final answer is correct and 0 otherwise.

$R_{fms}^k$ measures the quality of frame retrieval at the turn level, with a maximum reward of 0.5. At each intermediate turn $k$, the relevance of the retrieved frames $\mathcal{F}_k$ to the QA pair is quantified by the IoU with the ground-truth frames $\mathcal{G}$. The IoU score is tracked across turns, and a reward of 0.5 is assigned only if the current retrieval improves upon the best IoU achieved so far; otherwise, a penalty proportional to the IoU drop is applied. This design emphasizes marginal improvements

Figure 2: Illustration of Video-MTR's Multi-turn Reasoning Process, visualizing sampled frames, reasoning process, and model actions per turn. The ground-truth answer is highlighted in orange. The green timeline indicates the positions of sampled frames in the video, reflecting the model's frame selection strategy at each reasoning turn.

in the retrieved frame set, effectively preventing reward hacking through redundant frame selection while encouraging more efficient evidence gathering.

We also apply a formatting reward of $R_{\text{format}}^k = 0.1$ at each turn if the model's output conforms to the required format. The details of format and implementation are provided in Appendix A.2.

### 3.2.2 GOAL-GATED REWARD SHAPING

To ensure that intermediate actions contribute to the ultimate goal of video understanding, we introduce a goal-gated reward shaping mechanism. In this design, frame-retrieval rewards are granted only when the final answer is correct, ensuring that only retrieval operations leading to successful outcomes are reinforced. This couples retrieval and answering within the policy, rather than optimizing them in isolation. In our experiments, this setting proved critical. Without such constraints, since frame-retrieval actions can be issued multiple times, the model tended to prioritize optimizing retrieval actions to accumulate positive signals, while neglecting the primary objective of improving video understanding accuracy.

$$R(\tau) = \mathbf{1}_{\{R_{\text{acc}} > 0\}} \cdot \sum_{k=0}^{K-1} (R_{\text{fms}}^k + R_{\text{format}}^k) + R_{\text{acc}} + R_{\text{format}}^K$$

We aggregate the refined rewards into final reward-annotated trajectories, which then serve as training data for policy optimization.

### 3.3 REINFORCEMENT LEARNING

The standard RL objective function of the trajectory is defined as: $\max_{\pi_\theta} \ \mathbb{E}_{\tau \sim \pi_\theta} \big[ R(\tau) \big]$. We train the policy with Proximal Policy Optimization (PPO) and extend its default formulation to accommodate multi-turn reasoning. The multi-turn interactions trajectory is treated as an entire token sequence $\mathbf{z} = (z_0, z_1, \ldots, z_T)$. Instead of relying solely on sparse final-step feedback, the bi-level rewards are applied at every turn boundary and then propagated across all tokens $z_t$, enabling effective end-to-end learning. Specifically, two discount factors jointly shape the rewards during the calculation of token-level advantages $A_t^{GAE}$ :

- $\gamma_{\text{turn}}$: a cross-turn discount factor (0.95) applied to the accuracy reward $R_{acc}$, propagating the final answer signal back to earlier turns. At the boundary of turn $k$, the assigned reward is the original frame-retrieval reward of that turn plus a discounted accuracy term: $R_{\text{fms}}^k + \gamma_{\text{turn}}^{K-k} R_{acc}$.

| Model | Size | Frames | VideoMME Overall(w/o sub.) | MLVU Test | LongVideoBench Val | LVBench Overall | EgoSchema Subset |
|---|---|---|---|---|---|---|---|
| *Proprietary Models or Input Frame Budget: $> 256$ frames* | | | | | | | |
| GPT-4o (Hurst et al., 2024) | - | 0.5 fps / 384 | 71.9 | **54.9** | **66.7** | **48.9** | **72.2** |
| Gemini-1.5-Pro (Team et al., 2024) | - | 0.5 fps | **75.0** | - | 64.0 | 33.1 | 71.1 |
| DrVideo(GPT-4) (Ma et al., 2025) | - | 0.2/0.5 fps | 51.7 | - | - | - | 66.4 |
| Qwen2.5-VL-7B[†] (Bai et al., 2025) | 7B | 768 | 65.1 | - | 56.0 | 45.3 | 65.0 |
| VideoLLaMA2 (Cheng et al., 2024) | 8×7B | 8 | 47.9 | 45.6 | - | - | 53.3 |
| Video-CCAM (Fei et al., 2024) | 9B | 96 | 50.3 | 42.9 | 43.1 | - | - |
| LongVA (Zhang et al., 2024) | 7B | 128 / 256 | 52.6 | 41.1 | 47.8 | 37.9 | - |
| Video-XL (Shu et al., 2025) | 7B | 128 / 256 | 55.5 | 45.6 | 50.7 | - | - |
| VideoAgent (Wang et al., 2024b) | - | 87 | 56.0 | - | - | - | 60.2 |
| VideoMemAgent (Fan et al., 2024) | - | 72 | 57.4 | - | - | - | 62.8 |
| Video-LLaVA (Lin et al., 2023) | 7B | 8 | 39.9 | 30.7 | 39.1 | - | 36.8 |
| VideoChat2 (Li et al., 2024b) | 7B | 16 | 39.5 | 30.1 | 39.3 | - | - |
| LLaVA-OneVision (Li et al., 2024a) | 7B | 32 | 58.2 | - | 56.3 | - | 60.1 |
| Video-R1 (Feng et al., 2025) | 7B | 32 | 59.3 | 45.4 | - | 35.9 | 48.8 |
| Video-R1 (Feng et al., 2025) | 7B | 64 | 61.4 | 47.6 | - | 38.0 | 51.8 |
| Qwen2.5-VL-7B* (Bai et al., 2025) | 7B | 32 | 53.6 | 41.6 | 45.8 | 30.3 | 59.4 |
| Qwen2.5-VL-7B* (Bai et al., 2025) | 7B | 64 | 58.4 | 41.8 | 47.0 | 33.7 | 62.6 |
| Qwen2.5-VL-7B* (Bai et al., 2025) | 7B | 80 | 59.5 | 45.2 | 48.4 | 33.6 | 63.5 |
| Video-MTR | 7B | 32 | 59.0 | 48.4 | 52.3 | 38.2 | 62.4 |
| Video-MTR | 7B | 64 | 62.2 | 49.8 | 54.8 | 41.8 | 63.4 |
| Video-MTR (Ours) | 7B | 80 | **62.7** | **50.4** | **57.1** | **42.3** | **68.8** |

Table 1: Performance on mainstream long-video benchmarks. [†]: results reported in the original paper; *: results from our re-implementation/evaluation under different input settings. Best and second-best per category are **bolded** and underlined, respectively.

- $\gamma_{\text{token}}$: a within-turn discount factor (1.0) that propagates the turn boundary reward to tokens within the same turn.

The computed token-level advantages $A_t^{\text{GAE}}$ are then used in the standard PPO surrogate objective, ensuring that the sparse bi-level supervision signals jointly guide policy optimization. In practice, optimizing this objective presents two core challenges: (1) precisely estimating the intermediate frame-retrieval rewards; and (2) shifting a model originally biased toward single-turn reasoning into a multi-turn paradigm. We address these challenges with two strategies: a high-quality data curation pipeline that delivers fine-grained temporal supervision, and an exploration bootstrapping mechanism that incentivizes multi-turn retrieval behavior during early training.

**Data Curation** Computing turn-level frame-retrieval rewards requires temporally grounded annotations aligned with the problem, which most video-understanding datasets lack. A notable exception is NExT-GQA (Xiao et al., 2024) with 10.5K explicit grounding annotations. We retain instances with a relevant-segment ratio below 0.5 to enforce tighter temporal grounding, yielding roughly 5K high-quality samples. To scale and diversify training data, we additionally leverage video temporal grounding datasets such as QVHighlights (Lei et al., 2021), which provide precise temporal annotations for query-relevant segments. We adapt them to our QA-centric training by using GPT-4o (Hurst et al., 2024) to convert each query into a QA pair while preserving the original temporal alignment. To ensure quality, we apply a two-stage filter: (i) the LLM judges whether a query is suitable for QA conversion (discarding overly short or generic queries); (ii) we keep only instances with a relevant-segment ratio $< 0.5$. This produces nearly 3K QA-grounded samples from QVHighlights. This produces nearly 3K QA-grounded samples from QVHighlights. In total, we curate an 8K, compact yet supervision-rich set of temporally grounded examples. Departing from large-scale collection, we prioritize reward-signal fidelity over data volume, enabling efficient RL that attains competitive performance with far less data. We validate this in experiments by comparing efficiency and effectiveness against alternative approaches that rely on larger-scale data.

**Exploration Bootstrapping** During early rollouts, we observe that the pretrained MLLM rarely initiates evidence seeking. We omit supervised instruction tuning and introduce an adaptive exploration bonus: within each mini-batch, if the agent's frame–retrieval rate falls below a threshold, each retrieval action receives a small positive reward regardless of relevance; once retrievals become routine, the bonus is automatically disabled. This dynamic shaping bootstraps exploration, enabling pure RL to learn multi-turn evidence-seeking behavior.

# 4 EXPERIMENTS

## 4.1 IMPLEMENTATION DETAILS

Video-MTR is built upon the Qwen2.5-VL-7B and trained using the VAGEN framework, which supports multi-turn reinforcement learning. The policy is trained with PPO using a batch size of 32, an actor learning rate of $1 \times 10^{-6}$, and a critic learning rate of $1 \times 10^{-5}$.

**Number of Turns** We set the maximum number of turns $K_{\max}$ to 3, achieving a balanced compromise between accuracy and efficiency. A detailed examination, including quantitative comparisons under varying settings, is reported in the Appendix A.3.

**Input Frame Budget** Most LVD post-training methods operate with $\leq 128$ frames to align with training sequence lengths and manage computation. Given our practical resource constraints and to emphasize reasoning paradigm rather than raw capacity, we cap the input at 80 frames. Under the same budget, we compare: (i) a single-turn baseline with uniformly sampled frames; and (ii) our multi-turn framework that actively retrieves non-uniform subsets across turns, holding other factors fixed to isolate the effect of multi-turn reasoning. We evaluate budgets of 32, 64 and 80, and results consistently show that distributing frames over multiple retrieval–reasoning steps outperforms single-turn baseline. Concretely, the first turn uniformly samples half the budget, and each subsequent turn retrieves up to one quarter, ensuring the total never exceeds the frame budget.

## 4.2 BENCHMARKS

We select five representative long-form video benchmarks for comprehensive evaluation. Among them, VideoMME(Fu et al., 2025) is one of the most widely used benchmarks for general video understanding. To more closely target the challenges of long-form video reasoning, we further include MLVU(Zhou et al., 2024), LongVideoBench(Wu et al., 2024) and LVBench(Wang et al., 2024a), both featuring significantly extended video durations and complex task designs that rigorously test the capabilities and limitations of current MLLMs. Finally, we include the egocentric benchmark EgoSchema(Mangalam et al., 2023) of first-person human activities to evaluate the model's generalization across diverse scenarios.

## 4.3 PERFORMANCE OF LONG-FORM VIDEO UNDERSTANDING

### 4.3.1 MAIN RESULTS

We use objective questions across all benchmarks. The main results are summarized in Table 1. For long video understanding, achieving strong performance in prior work typically relies on either ultra-large proprietary models with hundreds of billions of parameters, or processing a substantial number of sampled frames, both of which are highly resource-intensive. For fairness, we report model size and input frame count alongside accuracy. Under comparable parameter and frame scales, Video-MTR shows clear advantages across all benchmarks. Notably, despite using only 7B parameters, Video-MTR achieves comparable performance on some of the *most challenging* long-video datasets, such as MLVU and LVBench, when compared to ultra-large proprietary models like GPT-4o and Gemini-1.5-Pro, which have significantly larger parameter sizes and more input frames. For example, on LVBench, Gemini-1.5-Pro processes $> 3000$ frames for 33.1% accuracy, whereas Video-MTR attains 42.3% with only 80 frames. Video-MTR with **80** input frames already achieves performance comparable to Qwen2.5-VL-7B with **768** frames across most of the datasets, and even outperforms it on EgoSchema (+3.8%) and LongVideoBench (+1.1%). We further analyze Video-MTR's advantages and summarize key findings below.

**Data-Efficient Supervision** Beyond accuracy, we compare training paradigms and data requirements across approaches in Table 2. For a strictly fair comparison, we only compare the data used during the fine-tuning stage for LVU. Most counterparts rely on hundreds of thousands to millions of supervised multimodal pairs, whereas Video-MTR is post-trained in a single RL stage with only 8K supervision-rich examples. Despite the drastic reduction in data scale, our model matches or even surpasses methods trained on vastly larger datasets across mainstream long-video benchmarks. To further validate this RL training paradigm, we applied the same procedure to Qwen2.5-VL-3B. Even with this smaller backbone, the model rapidly gained multi-turn reasoning capability, outperform-

ing its original single-turn baseline. Detailed results are provided in Appendix A.4. These findings show that the proposed paradigm is scalable and highly data-efficient. With just one to two training epochs, Video-MTR transforms an open-source MLLM from a single-turn to an iterative reasoner, offering a practical, cost-effective solution for long-video understanding.

**Benefits of Extended Frame Budgets** We compare performance under frame budgets of 32, 64 and 80, observing consistent gains across nearly all benchmarks. This trend holds for both Qwen2.5-VL-7B and Qwen2.5-VL-3B backbones, suggesting that extending models to handle longer video inputs is a promising avenue for future research.

### 4.3.2 CASE STUDY

Figure 2 illustrates Video-MTR's multi-turn reasoning on a 54-minute video for a single-detail query hinging on a critical plot point. In Turn 1, frames are uniformly sampled across the entire video. Noting that key evidence is missing, Video-MTR autonomously retrieves densely sampled segments semantically aligned with the query. In Turn 2, it re-examines the refined, query-relevant frames, extracts the required detail, and outputs the correct answer. This case shows how iterative retrieval and focused inspection overcome the limitations of uniform sampling in long videos.

### 4.4 ABLATION STUDY

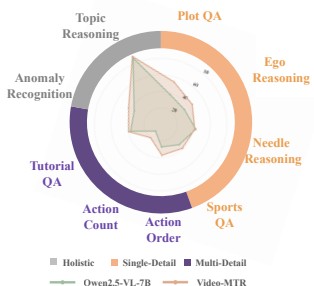

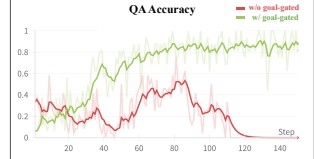 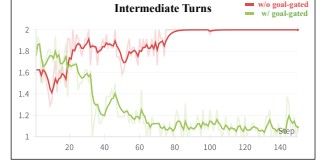

Figure 4: Reward hacking example. The red curve shows in the w/o goal-gated setting, the agent may simply accumulating more turns to increase reward, but with no corresponding gain in QA accuracy, whereas the green curve shows both increasing consistently.

Figure 3: Task Diagnosis.

| Method | Paradigm | Modalities | Volume |
|---|---|---|---|
| Video-CCAM | SFT | img/vid-text | 4.4M |
| VideoChat2 | SFT | img/vid-text | 2M |
| LongVA | SFT | img-text | 1.3M |
| Video-XL | SFT | img/vid-text | 257K |
| Video-R1 | SFT+RL (S) | img/vid-text | 260K |
| Video-MTR (Ours) | RL(M) | vid-text | 8K |

Table 2: Comparison of training paradigms, data modalities and volumes. (M)/(S) denote multi-turn and single-turn respectively.

| Model | Frames | VideoMME (w/o sub.) | | |
|---|---|---|---|---|
| | | Short | Medium | Long |
| Qwen2.5-VL-7B | 32 | 65.8 | 50.3 | 44.7 |
| Video-MTR | 32 | 70.4$^{+4.6}$ | 55.6$^{+5.3}$ | 51.0$^{+6.3}$ |
| Qwen2.5-VL-7B | 64 | 72.1 | 55.9 | 47.1 |
| Video-MTR | 64 | 72.8$^{+0.7}$ | 62.3$^{+6.4}$ | 51.4$^{+4.3}$ |
| Qwen2.5-VL-7B | 80 | 73.1 | 56.7 | 48.3 |
| Video-MTR (Ours) | 80 | 74.8$^{+1.7}$ | 60.6$^{+5.9}$ | 52.7$^{+4.4}$ |

Table 3: Comparisons of accuracy improvements across video durations.

We further investigate the contributions of several key components through detailed ablation studies.

### 4.4.1 ANALYSIS OF THE MULTI-TURN REASONING

We analyze the advantages of the proposed multi-turn reasoning framework over the conventional single-turn paradigm. Since Video-MTR is built on Qwen2.5-VL-7B, we compare directly against this base model to isolate performance gains. As multi-turn reasoning is expected to be particularly beneficial for complex tasks, we empirically assess its impact across diverse task types and video durations. **(1)Task types.** Using the MLVU benchmark, which categorizes evaluation tasks into three types: holistic tasks (global understanding of the entire video), single-detail tasks (focusing on one critical plot), and multi-detail tasks (requiring reasoning over multiple events), we observe distinct trends in Figure 3. For holistic tasks, typically lower in complexity, the base model achieves up to 72% accuracy, with Video-MTR providing a modest improvement of +3.8%. In contrast, detail-oriented tasks are substantially harder. The base model remains below 40% accuracy, while

| Ablation Setting | VideoMME (w/o sub.) | | | | LVBench |
|---|---|---|---|---|---|
| | Short | Medium | Long | Overall | Overall |
| Ours | 74.8 | 60.6 | 52.7 | 62.7 | 42.3 |
| Ours Multi-turn w/o Bi-Level Reward | 69.4 | 56.2 | 49.4 | 58.3 | 37.7 |
| Ours Single-turn | 68.8 | 54.8 | 47.9 | 57.2 | 35.3 |

Table 4: Ablation study. The first variant keeps the multi-turn paradigm but removes the bi-level reward. The second variant switches to a single-turn paradigm.

Video-MTR yields larger gains: +7.5% on single-detail and +8.1% on multi-detail. These results suggest a near-linear relationship between task complexity and the benefits of multi-turn reasoning. **(2)Video durations.** We further examine the impact of duration on VideoMME. We also observe a positive correlation between video length and performance gains. As shown in Table 3, under the 32-frame constraint, Video-MTR achieves accuracy improvements of +4.6% (Short), +5.3% (Medium), and +6.3% (Long) compared to Qwen2.5-VL-7B. Similarly, under the 64/80-frame constraint, the improvements for Medium and Long videos are notably higher than for Short videos.

To ensure a fair comparison, we further post-train Qwen2.5-VL-7B on the same data as Video-MTR. This yields our single-turn baseline, which processes the same number of uniformly sampled frames in a single forward pass. Compared with Video-MTR, it uses the same accuracy-based reward but removes multi-turn instructions from the prompts. Both models use identical optimization hyperparameters. Results for the single-turn baseline are reported in the third row of Table 4. While this single-turn variant yields modest improvements over Qwen2.5-VL-7B, it falls short when compared to Video-MTR, particularly on complex tasks in LVBench and long-form videos in VideoMME, consistent with our earlier analysis. This performance gap highlights the effectiveness of the multi-turn reasoning paradigm for complex inference.

### 4.4.2 EFFECTIVENESS OF BI-LEVEL REWARD

We evaluate the bi-level reward design against a multi-turn variant that omits this component, which removes turn-level supervision and relies solely on the final accuracy reward to guide the multi-turn behavior. As shown in Table 4, even with identical prompts and preserved multi-turn behavior, accuracy declines across benchmarks (including a significant 4.6% drop on LVBench). These findings highlight that, without intermediate supervision, relying solely on a final accuracy reward is insufficient to guide the model toward effective temporal localization, thereby limiting its reasoning capability.

### 4.4.3 NECESSITY OF GOAL-GATED REWARD SHAPING

To assess the effectiveness of our goal-gated reward shaping in mitigating reward hacking, we compare Video-MTR with an ablated variant that removes this mechanism and instead receives unconditioned turn-level rewards. Figure 4 shows the resulting failure mode that emerges early in training: during training, the ablated agent inflates reward by repeatedly retrieving frames with more turns rather than answering correctly. By contrast, the goal-gated model keeps reward and task success closely aligned. These results confirm that goal-gated shaping is crucial for preventing superficial reward exploitation and preserving genuine video understanding capability.

## 5 CONCLUSION

We present Video-MTR, a reinforced multi-turn reasoning framework for long-form video understanding. To the best of our knowledge, it is the first work to integrate end-to-end reinforcement learning with explicit multi-turn reasoning in this domain. At the core of the framework is a gated bi-level reward mechanism, designed to incentivize both relevant frame retrieval and step-by-step reasoning. Extensive experiments on VideoMME, MLVU, LongVideoBench, LVBench and EgoSchema demonstrate that Video-MTR achieves strong and robust performance across diverse task types and varying temporal lengths. Notably, the framework exhibits excellent temporal scalability, yielding higher gains as video duration increases, highlighting its particular advantage in extra-long video understanding. Future work includes extending the framework to even longer videos and more complex reasoning tasks, pushing the boundaries of long-video understanding.

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

## A  TRAINING DETAILS

### A.1  PROMPT DESIGN

This section details our prompt design and provides an illustrative example in Figure 6. To incentivize multi-turn reasoning, we craft an instruction template that guides the MLLM to follow a predefined interaction protocol. The prompt is multimodal: visual tokens corresponding to frames observed in the current turn are inserted immediately after their textual description. We then append a format template that constrains the model's output to a structured schema. We define two actions per turn: (i) `answer`, which outputs only the single option letter; and (ii) `retrieve`, which outputs `start_frame` and `end_frame`. In each turn, the model is explicitly required to first provide a brief rationale and then emit the action in the specified format.

### A.2  FRAME RETRIEVAL PROTOCOL

We next describe the frame-retrieval format and implementation. At preprocessing, we uniformly subsample up to $M$ frames from each video to form a candidate pool $\mathcal{F}_{all}$ and index them accordingly; in our implementation we set $M = 128$, which worked well empirically. In the frame budget settings of 32, the agent receives a sparse overview of 16 uniformly spaced frames in the initial turn. In subsequent turns, the agent may issue a retrieval action that selects a temporal interval by outputting `start_frame` and `end_frame` ($\mathcal{F}_{all}$). The environment then returns frames from this interval at an appropriate stride, capped at most 8 frames. This procedure allows the model to iteratively focus on key segments by selecting targeted subsets of frames.

## A.3 ANALYSIS OF TURN LIMIT

Although multi-turn reasoning improves accuracy through iterative evidence gathering, it requires multiple forward passes, leading to increased inference latency. This creates a fundamental trade-off between efficiency and performance. To quantify it, we conducted controlled experiments under different maximum-turn settings ($K_{max}$). All experiments are performed with the Qwen2.5-VL-7B backbone, using a fixed total input frame budget of 32 frames to ensure comparability across settings. The model is evaluated on benchmark datasets with identical training and inference conditions, while varying only the maximum number of turns allowed during training. Results in Table 5 show that while additional turns improve accuracy, the gains diminish beyond a certain point, whereas latency grows nearly linearly. Based on this analysis, we set the maximum number of turns $K_{max}$ to 3 and retain the last 2 turns as context, achieving a balanced compromise between accuracy and efficiency.

| Max Turns $K_{max}$ | Avg. Turns Used | Accuracy (M-AVG, %) | | Latency (ms) |
|---|---|---|---|---|
| | | VideoMME | MLVU | |
| 1 | 1.0 | 54.8 | 42.6 | 194.4 |
| 2 | 1.6 | 57.9 | 43.1 | 312.2 |
| 3 | 2.2 | 59.0 | 48.4 | 427.2 |
| 5 | 3.2 | 60.7 | 47.4 | 622.8 |

Table 5: Accuracy on VideoMME and MLVU, latency, and average number of turns actually used under different maximum-turn settings $K_{max}$.

## A.4 ADDITIONAL RESULTS ON QWEN2.5-VL-3B

To further verify the generality of our end-to-end reinforcement learning training paradigm, we applied the same procedure to Qwen2.5-VL-3B. Despite its smaller capacity compared to Qwen2.5-VL-7B, the model rapidly acquired multi-turn reasoning ability and consistently outperformed its single-turn baseline. These results in Table 6 demonstrate that the proposed framework is not only effective for larger backbones but also generalizes well to lighter models under limited resources.

## A.5 IMPLEMENTATION OF EXPLORATION BOOTSTRAPPING

To address the lack of proactive evidence seeking in early training, we introduce an adaptive exploration bonus that bootstraps multi-turn retrieval. We compute statistics at the mini-batch level (batch size = 32) and use a two-stage schedule. For each mini-batch, if the retrieval rate (fraction of turns issuing a `retrieve` action) falls below a stage-specific threshold, we add a fixed bonus to every retrieval action in that batch, irrespective of frame relevance.

- Stage I (cold start): threshold = 0.1, bonus = +1.0.
- Stage II (bootstrapping): threshold = 0.5, bonus = +0.5.

Once the retrieval rate remains above the Stage-II threshold for several consecutive mini-batches, the bonus is disabled. As shown in Figure 5, this dynamic shaping reliably kick-starts and sustains multi-turn evidence-seeking behavior under pure RL.

| Model | Frames | VideoMME (w/o sub.) | | MLVU | LVBench | EgoSchema |
|---|---|---|---|---|---|---|
| | | Long | Overall | Test | Overall | Subset |
| **3B Models** | | | | | | |
| Qwen2.5-VL-3B | 32 | 43.6 | 51.5 | 41.2 | 31.2 | 57.4 |
| Video-MTR (3B) | 32 | 46.8$^{+3.2}$ | 52.5$^{+1.0}$ | 42.4$^{+1.2}$ | 36.1$^{+4.9}$ | 59.5$^{+2.1}$ |
| Qwen2.5-VL-3B | 64 | 45.9 | 54.0 | 43.4 | 34.7 | 59.4 |
| Video-MTR (3B) | 64 | 45.4$^{-0.5}$ | 54.7$^{+0.7}$ | 47.1$^{+3.7}$ | 36.7$^{+2.0}$ | 64.2$^{+4.8}$ |

Table 6: Comparison of single-turn and multi-turn settings on Qwen2.5-VL-3B. The multi-turn framework consistently improves accuracy.

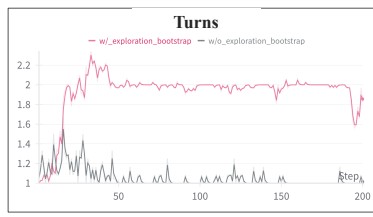 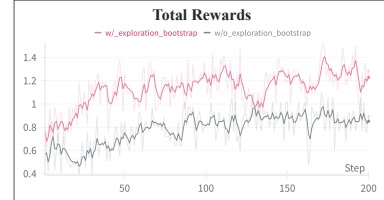

Figure 5: Exploration bootstrapping enables multi-turn behavior. With the bonus (pink), rewards grow as multi-turn retrieval is maintained; without it (gray), the policy stabilizes at single-turn reasoning.

---

**System:** conversation between User and Assistant. The user asks a question, and the Assistant solves it. You are an assistant in analyzing videos. Your will be given a video and a question. Goal: Answer the question correctly with no more than 3 turns.

**User:** Turn 1. Now you are given 16 selected frames from the video, with frame_idx_list: [ 0 4 8 12 16 21 25 29 33 37 42 46 50 54 58 63].

Frames: frame_idx:0, `<image_0>`,frame_idx:4, `<image_4>`,frame_idx:8, `<image_8>`, ..., frame_idx:54, `<image_54>`,frame_idx:58, `<image_58>`,frame_idx:63, `<image_63>`

Answer the following problem based on the frames: What is the event in the video? Options: ["A. Women's synchronized diving", 'B. Individual diving', 'C. Swimming' "D. Men's synchronized diving", 'E. Swimming relay race', 'F. Synchronized swimming'] You can choose to retrieve more frames or provide your answer.

Format Template:

`<think>`...`</think><answer>`...`</answer>` or `<think>`...`</think><retrieve>`...`</retrieve>`

Please think about this question as if you were a human pondering deeply. Engage in an internal dialogue using expressions such as 'let me think', 'wait', 'Hmm', 'oh, I see', 'let's break it down', etc, or other natural language thought expressions. It's encouraged to include self-reflection or verification in the reasoning process. Provide your detailed reasoning between the `<think>` and `</think>` tags. If you have enough information, Please provide only the single option letter (e.g., A, B, C, D, etc.) within the `<answer>` `</answer>` tags.

If you lack some information, think about the most relevant frame index range of the information you need, then you can retrieve dense frames in the range by sending a retrieve request by `<retrieve>` start_frame, end_frame `</retrieve>`.

IMPORTANT: start_frame and end_frame must be integers smaller than 63.

**Video-MTR:** `<think>`I need to identify the event and verify if it is men's synchronized diving. Let's look for any indication that this is a synchronized diving event and verify if it is men's synchronized diving.`</think><retrive>`12,33`</retrive>`

------------------------------------------------------------

**User:** Turn 2. Now you are given 8 selected frames from the video, with frame_idx_list: [12 15 18 21 24 27 30 33].

Frames: frame_idx:12, `<image_12>`,frame_idx:15, `<image_15>`,frame_idx:27, `<image_27>`, ..., frame_idx:30, `<image_30>`,frame_idx:33, `<image_33>`,

Answer the following problem based on the frames: What is the event in the video?

...

If you lack some information, think about the most relevant frame index range of the information you need, then you can retrieve dense frames in the range by sending a retrieve request by `<retrieve>` start_frame, end_frame `</retrieve>`.

**Video-MTR:** `<think>`I need to verify if this is a synchronized diving event and check if it is men's synchronized diving. The frames show two divers performing a synchronized dive, which indicates a synchronized diving event. `</think><answer>`D`</answer>`

---

Figure 6: An example of Video-MTR with interactions of 2 turns. **`<image_n>`** will be replaced with the visual tokens of the specific frames.

## B   DATASETS

This section details the construction and statistics of our temporally grounded supervision dataset for reinforcement learning (RL) training. The dataset comprises two components: one curated from a video-understanding dataset NExT-GQA and one adapted from a video temporal grounding dataset QVHighlights:

**Goal:** Given a declarative sentence to serve as a query for retrieving relevant video segments, generate a multiple choice question.
Follow these rules:
1. Suitability Check: Return False if the sentence is too short or Lacks distinctive details for discriminative options. Else, return True and proceed.
2. Question Format: Use one of these interrogatives: Where, How, Why, What, When, Who
3. Options: Derive one correct answer and three incorrect answers from the sentence.
4. Answer: The correct answer to the question.
**Format**
{
   "suitable": bool, # True/False
   "question": str, # MCQ text (if suitable)
   "options": list,
   "answer": str # Correct option
}
**Examples**
- Sentence:A man in white shirt discusses the right to have and carry firearms.
- Output:{
   "suitable": True
   "question": What is the man in a white shirt discussing?
   "options": ["A. The war happens in Europe.", "B. The recent massacre in the US.", "C. The right to have and carry firearms.", "D. The recent crime in the US."]
   "answer": C
}
- Sentence: Woman holds her shopping bags.
- Output:{
   "suitable": False
   "question":""
   "options": ""
   "answer": ""
}
_ _ _ _ _ _ _ _ _ _ _ _ _ _ _ _ _ _ _ _ _ _ _ _ _ _ _ _ _ _ _ _ _ _ _

**QA Converted Examples**

   "- query":"Asian chef with dyed pink hair cooks food."
   "- question": "What is the Asian chef with dyed pink hair doing?"
   "- options": ["A. Preparing ingredients", "B. Serving customers", "C. Cleaning the kitchen", "D. Cooking food"],
   "- answer" : "D"

   "- query": "Two people from the same show interview a man at his house."
   "- question": "Where do two people from the same show interview a man?"
   "- options": ["A. At his house", "B. In a studio", "C. Outside", "D. In an office"]
   "- answer" : "A"

Figure 7: The GPT-4o prompt template for converting declarative queries into multiple-choice QA pairs with suitability check, options generation, and converted QA examples.

- NExT-GQA Starting from 10.5K explicit temporal grounding annotations (consolidated into 8.9K QA pairs), we retain instances with a relevant-segment ratio $< 0.5$ and video duration $> 30s$, yielding $\sim$5K high-quality samples.

- QVHighlights We use GPT-4o to convert each original query into a QA pair aligned with its temporal annotations, and apply a two-stage quality filter: (i) discriminative-adequacy screening; and (ii) relevant-segment ratio $< 0.5$ and video duration $> 30s$, resulting in $\sim$3K QA-grounded samples.

In total, we obtain 8K training instances that are compact yet supervision-dense. Table 7 reports per-source composition and retained counts at each step to facilitate reproduction and extension. Figure 7 illustrates the GPT-4o prompt design for rewriting and provides before/after examples.

| Source | Pre Filter | QA Converted | Post Filter |
|--------|-----------|--------------|-------------|
| NExT-GQA | 8.9K | - | 4.9K |
| QVHighlights | 7.2K | 3.5K | 3.0K |

Table 7: Dataset composition and filtering statistics. Counts denote thousands of samples. NExT-GQA is directly used as QA pairs.

## C  CASE STUDIES

We present additional case studies drawn from three evaluation benchmarks—VideoMME (Fu et al., 2025), MLVU (Zhou et al., 2024), and EgoSchema (Mangalam et al., 2023) to give a comprehensive picture of Video-MTR's multi-round reasoning process; these examples include both successes and failures.

### C.1  SUCCESSFUL CASES

From each dataset we randomly selected one correctly solved example. As illustrated in Figure 8, all three examples exhibit a consistent evidence-seeking pattern with the following characteristics: (i) an initial global pass over the video produces a tentative hypothesis that roughly answers the question; (ii) the model then proposes a targeted temporal segment for closer inspection to obtain discriminative evidence; and (iii) after observing this segment, the model updates or confirms the hypothesis and outputs the final answer.

**Case A (role identification).** The query asks for the identities of two people. After the first pass, Video-MTR hypothesizes that the pair may be a teacher and a student based on coarse contextual cues from the full video. It then narrows attention to their interaction segment for verification. In that focused clip, the person in a white shirt is seen giving instructions, and the standing man in a black shirt follows the instructions and plays the instrument. This instructional exchange provides role-asymmetric signals: directive speech acts, demonstrative gestures, and action–response ordering, yielding temporally grounded, discriminative evidence that confirms the teacher–student hypothesis.

**Case B (event recognition).** The question asks which event is shown, with candidates including individual/synchronized diving, swimming, relay, and synchronized swimming. After a global pass, Video-MTR sets a verification subgoal: to confirm synchronized diving—and proposes a discriminative interval for inspection. Focusing on this clip, the model observes two divers executing the same dive with mirrored body alignment, thereby ruling out individual diving and all swimming events. The model confirms the hypothesis and outputs (D) Men's synchronized diving.

**Case C (goal reasoning).** The query seeks a concise account of C's objective and decisions. After a first pass, Video-MTR hypothesizes that C is choosing what to wear and proposes a targeted interval for verification. In this segment, C looks at various clothes, picks them up, and appears to be deciding what to wear, with no behaviors indicative of folding, packing, ironing, or washing. The model confirms the hypothesis and outputs (C) deciding what clothes to wear.

### C.2  ERROR ANALYSIS AND LIMITATIONS

We also examine failure cases to diagnose error sources and outline potential remedies. Two representative cases, one involving multi-detail reasoning and the other requiring fine-grained perception are illustrated in Figure 9.

**Case A (Action Order).** This example falls under the action-order category, a multi-detail task requiring inspection of multiple, disjoint segments. In Rounds 1–2 the sampled frames do not cover all events referenced by the options; nevertheless, the model commits to a prediction, exhibiting hallucination under insufficient evidence. More retrieval rounds are needed to reach a reliable decision. A likely cause is a training-distribution bias: in our data, one to three rounds typically suffice to locate relevant frames and answer correctly, which encourages early stopping even when evidence is incomplete. A straightforward remedy is to expand the curriculum with more sequences that demand four to six retrieval rounds and span widely separated events, prompting the model to keep searching until each candidate answer has been either supported or ruled out.

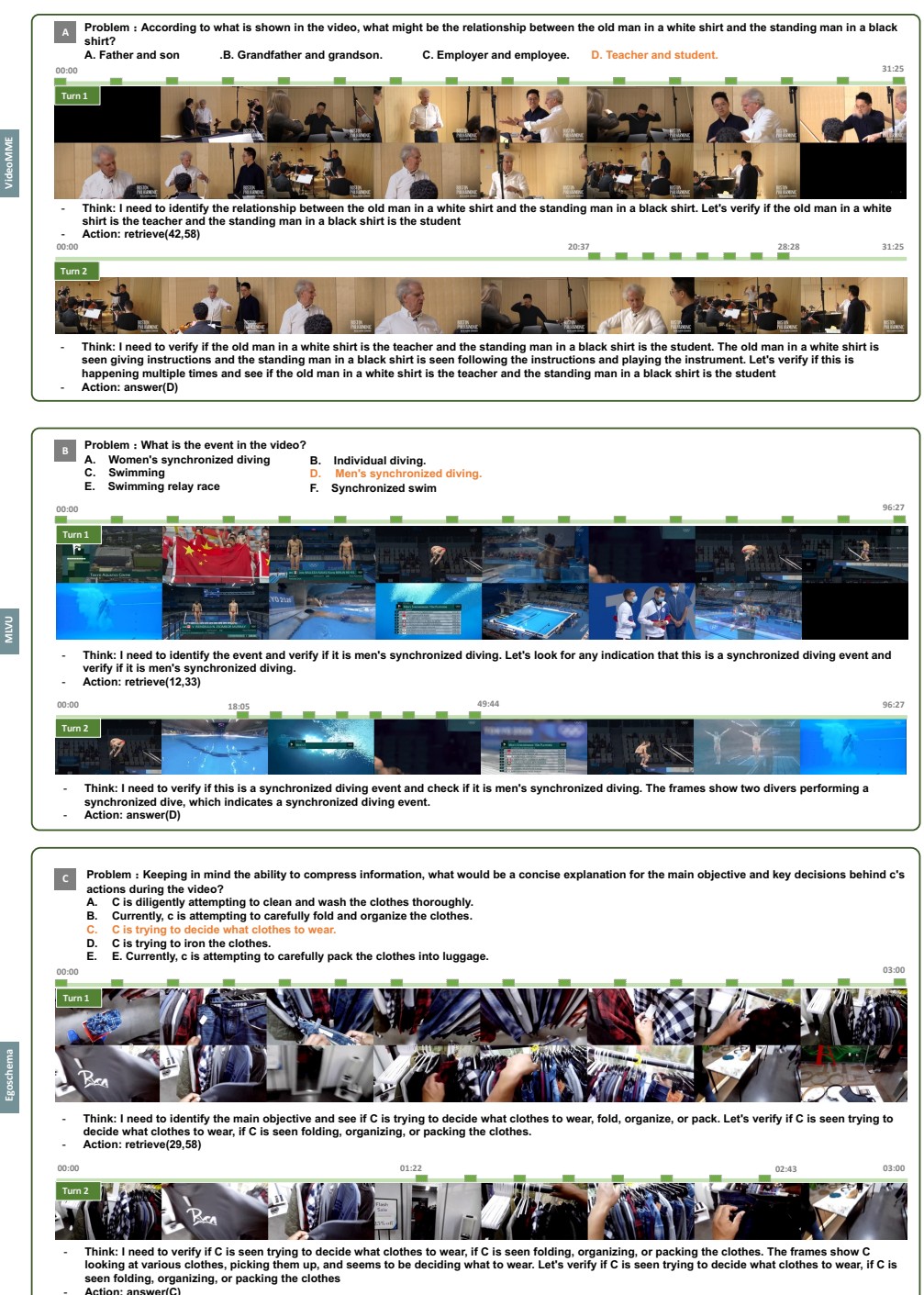

Figure 8: Representative success cases from (A) VideoMME, (B) MLVU, and (C) EgoSchema. The ground-truth answer is highlighted in orange. The green timeline indicates the positions of sampled frames in the video.

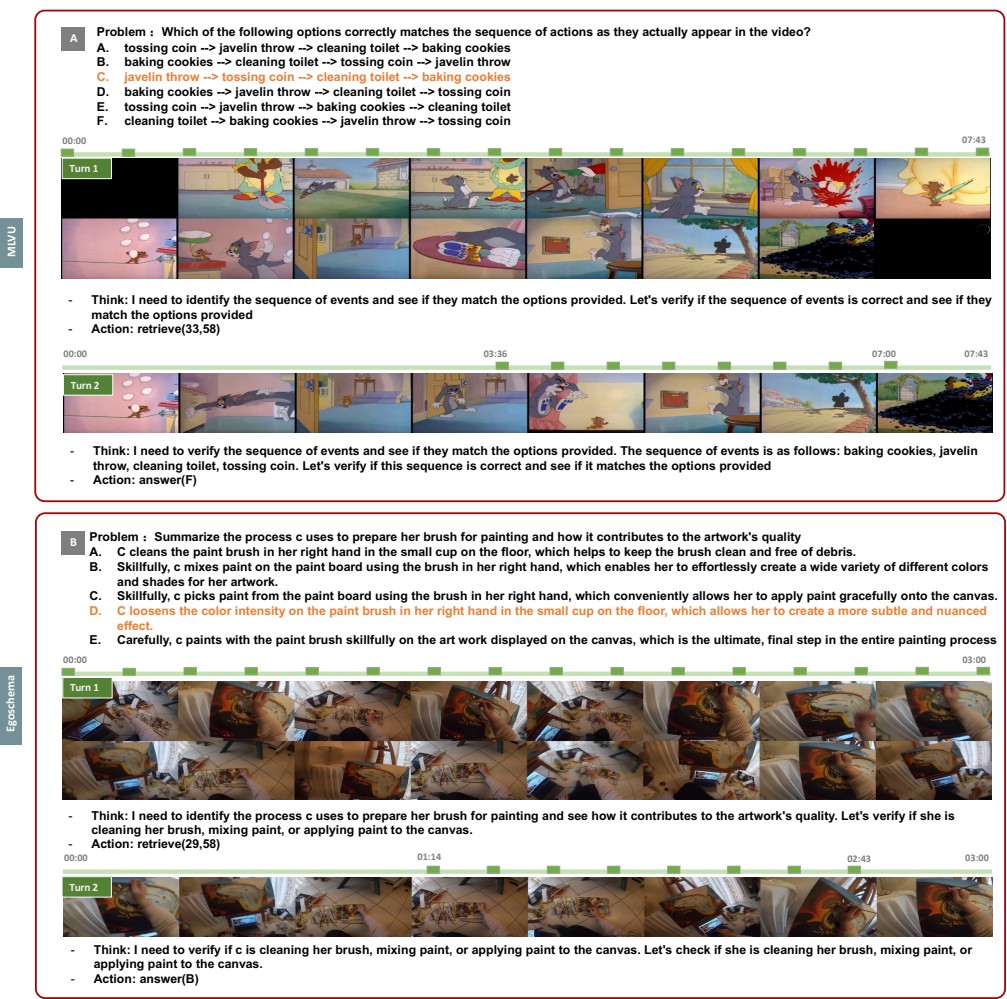

Figure 9: Representative failure cases: (A) action-order reasoning error and (B) fine-grained procedural misrecognition.The ground-truth answer is highlighted in orange. The green timeline indicates the positions of sampled frames in the video.

**Case B (Fine-grained Procedural Reasoning).** This task requires interpreting micro-actions (e.g., dipping or swishing in a cup versus mixing on a palette) and linking them causally to paint subtlety. Under the current frame-processing pipeline, which must accommodate long temporal sequences, the spatial resolution is kept relatively coarse; as a result, these discriminative cues are likely to appear heavily blurred. To address this limitation, the retrieval-and-reasoning loop at the frame-selection level could be augmented with a hierarchical temporal-to-spatial reasoning mechanism: once a relevant frame segment is identified, the system would crop the corresponding frames and re-analyse high-resolution regions of interest, enabling direct verification of micro-movements before any answer is produced.

These failure cases several structural weaknesses that limit the current version of Video-MTR in complex scenarios. Together, these issues indicate that Video-MTR needs deeper temporal search policies, hierarchical zoom-in vision modules to handle multi-event reasoning and fine-grained perception reliably.

| Model | Params | Backbone (LLM) | Post-train Data | Frames / fps |
|---|---|---|---|---|
| GPT-4o (Hurst et al., 2024) | – | GPT-4o (proprietary) | – | 0.5 fps / 384 |
| Gemini-1.5-Pro (Team et al., 2024) | – | Gemini (proprietary) | – | 0.5 fps |
| DrVideo (GPT-4) (Ma et al., 2025) | – | GPT-4 (proprietary) | – | 0.2 / 0.5 fps |
| Qwen2.5-VL-7B[†] (Bai et al., 2025) | 7B | Qwen2.5-VL-7B | – | 768 |
| VideoLLaMA2 (Cheng et al., 2024) | 8×7B | Mixtral-8x7B-Instruct | 1.35M | 8 |
| Video-CCAM (Fei et al., 2024) | 9B | Yi-1.5-9B-Chat | 4.4M | 96 |
| LongVA (Zhang et al., 2024) | 7B | Qwen2-7B-Instruct | 760K | 128 / 256 |
| Video-XL (Shu et al., 2025) | 7B | Qwen2-7B | 257K | 128 / 256 |
| VideoAgent (Wang et al., 2024b) | – | GPT-4 (proprietary) | – | 87 |
| VideoMemAgent (Fan et al., 2024) | – | GPT-4 (proprietary) | – | 72 |
| Video-LLaVA (Lin et al., 2023) | 7B | Vicuna-7B-v1.5 | 765K | 8 |
| VideoChat2 (Li et al., 2024b) | 7B | Vicuna-7B-v0 | 2.0M | 16 |
| LLaVA-OneVision (Li et al., 2024a) | 7B | Qwen-2-7B | 4.8M | 32 |
| Video-R1 (Feng et al., 2025) | 7B | Qwen2.5-VL-7B | 260K | 32 / 64 |
| **Video-MTR (Ours)** | 7B | Qwen2.5-VL-7B | **8K** | **32 / 64 / 80** |

Table 8: Summary of compared baseline models, their backbones, frame budgets, and post-training data scale.

## D MORE COMPARISONS

To give a more comprehensive comparison: in addition to the original parameter size and frame budget Table 1, we now also summarize, for each baseline, its backbone LLM and post-training data scale. Regarding the backbone comparison, Table 8 shows that our setting is fair across different implementation choices: Video-MTR shares the exact same 7B Qwen2.5-VL-7B backbone with Video-R1 and uses the same 7B Qwen2 family as LongVA and Video-XL, yet achieves superior performance while being trained on significantly less data (only an 8K long-video QA corpus). In contrast, many strong baselines rely on proprietary GPT-4/Gemini backbones or web-scale multi-modal data. For the post-training data, to ensure a fair comparison, we compare only the data used in the post-training stage (instruction tuning or RL), rather than the full pre-training corpora. For this reason, we do not list the massive datasets used to build GPT-4/Gemini or the Qwen2.5-VL-7B backbone itself. Most counterparts rely on hundreds of thousands to millions of supervised multi-modal pairs, whereas Video-MTR is post-trained in a single RL stage with only 8K supervision-rich examples, clearly highlights the strong data efficiency of our framework.

## E FUTURE WORK

Although Video-MTR demonstrates strong reasoning performance on current long-form benchmarks, ample room for improvement remains when tackling more challenging queries and much longer videos. Future work should therefore advance the multi-round framework on two fronts: (i) lengthen the dialogue loop to support deeper chains of reasoning that solve multi-stage tasks, and (ii) incorporate a hierarchical temporal-to-spatial strategy that begins with coarse video sweeps and adaptively zooms into high-resolution frame crops, thereby securing reliable evidence at both event-level and micro-action scales.

