# OpenReview forum: "Video-MTR: Reinforced Multi-Turn Reasoning for Long Video Understanding"
_ICLR.cc/2026/Conference — Submitted to ICLR 2026_

### Official Review · Reviewer_JQqn · 2025-10-15

**Soundness:** 3
**Presentation:** 3
**Contribution:** 2
**Rating:** 4
**Confidence:** 4

**Summary:**

Video-MTR is a reinforced multi-turn reasoning framework for long-form video understanding, addressing challenges like long-range temporal dependencies and multi-event complexity. Unlike existing methods (static single-turn reasoning or external VLM-reliant agentic paradigms), it enables iterative key segment selection and end-to-end training via reinforcement learning (RL). Built on Qwen2.5-VL-7B, it uses a gated bi-level reward system (trajectory-level: answer correctness; turn-level: frame-query relevance via IoU) to guide multi-turn reasoning. It starts with uniform frame sampling, then retrieves relevant segments iteratively until confident or reaching a 3-turn limit. The proposed model is trained on only 8K curated samples (from NExT-GQA and QVHighlights), far fewer than existing methods (256K–4.4M).
Experimental Results: Outperforms baselines on 4 benchmarks (VideoMME: 62.2%, MLVU: 49.8%, LVBench: 41.8%, EgoSchema: 63.4%) with 7B parameters and ≤64 frames, matching large proprietary models (e.g., Gemini-1.5-Pro) with fewer resources.

**Strengths:**

Strong Long-Video Reasoning: Multi-turn iteration avoids missing critical info in long videos, with larger accuracy gains for longer videos (+6.3% on long vs. +4.6% on short in VideoMME) and complex tasks (+8.1% on multi-detail tasks in MLVU).

High Efficiency:
Data-efficient (8K samples vs. hundreds of thousands/millions).
Compute-efficient (7B parameters, ≤64 frames) vs. large models or high-frame-budget methods.
Balanced latency (427.2 ms at 3 turns) vs. accuracy.

End-to-End & Tool-Independent: Eliminates external VLMs, avoiding heterogeneous component complexity and enabling unified optimization of segment selection and question comprehension.

Robust & Generalizable: Performs consistently across benchmarks; works for smaller models (Qwen2.5-VL-3B) with accuracy gains.

**Weaknesses:**

1. The idea of first grounding/localization and then answering questions is not novel.

2 .Limitations in Complex Tasks: Struggles with multi-event (e.g., action-order) tasks (early stopping due to training bias) and fine-grained perception (coarse frames blur micro-actions like "brush dipping vs. mixing").

3. Turn-level rewards need high-quality temporal annotations, making scaling to new domains costly.

**Questions:**

No

---

> ### Author Response · Authors · 2025-11-20
>
> Dear Reviewer JQqn，
>
> We thank the reviewer for reviewing our manuscript and raising several specific points for clarification. We are pleased that the reviewer acknowledged the high efficiency and strong generalization of our method. Below, we address the reviewer’s three main concerns regarding novelty, limitations in complex tasks, and scaling cost.
>
> **Q1:The idea of first grounding/localization and then answering questions is not novel.**
>
> **A1**: Thanks for the comment. As we discuss in the introduction, there already exist works that implement the grounding-and-answering structure using modular, heuristic agents, and rely on extra modules for individual components (retrievers, planners, tools). However, they suffer from two main limitations: (i) they typically sit on top of powerful pretrained models (e.g., GPT-4o, Gemini) and adopt sub-optimal tool-usage strategies without end-to-end policy learning; and (ii) they introduce high system complexity due to the reliance on multiple external components. We argue that training a **single model end-to-end, while jointly optimizing both retrieval and answering** under a unified RL objective, can avoid these issues and yield a more robust and learnable multi-turn policy. Therefore, we propose Video-MTR, which formulates multi-turn grounding + QA as a single RL problem and optimizes a unified policy end-to-end on top of an open-source MLLM backbone. To the best of our knowledge, Video-MTR is the first reinforced multi-turn reasoning framework for long video understanding (LVD) with end-to-end policy training; in this sense, it constitutes a novel framework for long-video reasoning beyond existing modular agent designs.

---

> ### Author Response · Authors · 2025-11-20
>
> **Q2: Limitations in Complex Tasks: Struggles with multi-event (e.g., action-order) tasks (early stopping due to training bias) and fine-grained perception (coarse frames blur micro-actions like "brush dipping vs. mixing").**
>
> **A2**: Thank you for the comment. The two complex task types you mention, multi-event reasoning and fine-grained perception are exactly the failure modes we deliberately analyze in **Appendix C.2 (Error Analysis and Limitations)** to provide a more comprehensive diagnosis of Video-MTR. We would like to further clarify two points:
>
> First, our main results in Sec. 4.4.1 analyze the effect of task complexity and video duration on Video-MTR. The results show that Video-MTR not only substantially improves over the Qwen2.5-VL-7B backbone across diverse task types and video lengths, but, more importantly, that the **gains from multi-turn reasoning grow roughly linearly with task complexity and video duration**. In other words, while we are transparent about hard corner cases, the quantitative analysis demonstrates that our framework already brings significant benefits on challenging long-video tasks compared to existing baselines.
> Second, these challenging cases represent **open problems for current LVD systems**, and all models struggle on them. Our analysis is intended to diagnose structural weaknesses and to propose concrete directions for future improvement. Based on these observations, we outline potential remedies within the Video-MTR framework, including training deeper temporal search policies and adopting hierarchical zoom-in vision modules to better handle multi-event reasoning and fine-grained perception.

---

> ### Author Response · Authors · 2025-11-20
>
> **Q3: Turn-level rewards need high-quality temporal annotations, making scaling to new domains costly.**
>
> **A3**: Thank you for the comment. In fact, our framework is explicitly designed to be efficient, both **data-efficient** in terms of annotated supervision and **computationally efficient in training**. As you also noted in the Strengths (“High Efficiency: Data-efficient (8K samples vs. hundreds of thousands/millions)”), most long-video baselines require hundreds of thousands to millions of post-training examples, whereas Video-MTR, although it uses temporal annotations, is post-trained on **only 8K** supervision-rich long-video QA episodes. The additional temporal labeling cost is modest compared to existing large-scale instruction-tuning or RL recipes and, in practice, is far from being the bottleneck.
>
> Moreover, with this small-scale dataset, training is also highly efficient: **1–2 epochs of RL training** are sufficient for the model to acquire effective multi-turn reasoning behavior. In addition, the RL-based training confers strong generalization to new domains, our experiments with **strictly out-of-domain evaluation** show that Video-MTR learns a generalizable retrieval policy rather than overfitting to a single dataset. Consequently, the framework can be scaled to new domains while still retaining robust performance.

---

> ### Author Response · Authors · 2025-11-24
>
> Dear Reviewer  JQqn,
>
> We have submitted a revised version of the manuscript along with detailed responses to your comments. Could you please have a look at our response? We would greatly appreciate any further feedback and are looking forward to your reply.
>
> Authors of Video-MTR

---

> > ### Comment · Reviewer_JQqn · 2025-11-28
> > **Thanks for the rebuttal**
> >
> > Thank the authors for the rebuttal. Most of my concerns are addressed, I will increase my score to 6.

---

> > > ### Author Response · Authors · 2025-11-28
> > >
> > > Thank you very much for your positive assessment of our work. We sincerely appreciate your decision to raise the score.

---

### Official Review · Reviewer_R4Fn · 2025-10-31

**Soundness:** 3
**Presentation:** 2
**Contribution:** 3
**Rating:** 4
**Confidence:** 5

**Summary:**

This paper proposes Video-MTR, a reinforced multi-turn reasoning framework designed to enable iterative key video segment selection and question comprehension. Video-MTR performs reasoning in multiple turns, selecting video segments progressively based on the evolving understanding of previously processed segments and the current question. Extensive experiments on benchmarks like VideoMME, MLVU, LVBench, and EgoSchema demonstrate that VideoMTR outperforms existing methods in both accuracy and efficiency, advancing the state-of-the-art in long video understanding.

**Strengths:**

1. As claimed by the authors, this could be the first attempt to incorporate multi-turn reasoning in the context of long video understanding.
2. The proposed method can adaptively select important frames for the question.
3. Experiments on multiple long-video benchmarks show that Video-MTR outperforms its backbone Qwen2.5-VL with the same number of frames.

**Weaknesses:**

1. Qwen2.5-VL can support up to 768 frames and outperforms the proposed Video-MTR with the input of 64 frames. More experiments should be conducted to investigate whether Video-MTR can outperform Qwen2.5-VL with more frames.
2. It's weired that the QA accuracy in Figure 4 exceed 1. Some explanations should be provided.
3. The information of compared baseline models is not given, especially their backbone models.
4. While introducing multi-turn reasoning, the efficiency compared to baselines should be analyzed.
5. The training framework is complicated with various tricks, which may be unstable in other scenarios.

**Questions:**

Please reply to Weaknesses.

---

> ### Author Response · Authors · 2025-11-20
>
> Dear Reviewer R4Fn,
>
> We sincerely thank you for the thorough reviews and constructive feedback. We now present additional experiments and clarification regarding your concerns below.
>
> **Q1 : Video-MTR vs Qwen2.5-VL with larger input budget**
> > Qwen2.5-VL can support up to 768 frames and outperforms the proposed Video-MTR with the input of 64 frames. More experiments should be conducted to investigate whether Video-MTR can outperform Qwen2.5-VL with more frames.
>
> **A1** : We appreciate the reviewer’s suggestion to evaluate Video-MTR with more frames. However, given our practical resource constraints, training Video-MTR with hundreds of frames is currently infeasible for us.
> Within this constrain, We have tried to extend our experiments from 64 to 80 input frames, and, for  a more comprehensive comparison with Qwen2.5-VL-7B, we also include an additional commonly used long-video benchmark, LongVideoBench[1] that is also reported in the Qwen2.5-VL technical report.
> The new results show that increasing the input to 80 frames further improves performance: **Video-MTR (80 frames)** already achieves performance comparable to **Qwen2.5-VL-7B with 768** frames across five datasets, and even outperforms it on **EgoSchema (+3.8%) and LongVideoBench (+1.1%)**. As summarized in Table 1, we observe: (i) under matched frame budgets (32 / 64 / 80), Video-MTR consistently outperforms its backbone Qwen2.5-VL-7B, demonstrating the effectiveness of multi-turn reasoning; and (ii) more importantly, as the frame budget increases, Video-MTR exhibits a clear trend where **larger input frame numbers lead to steadily improved performance**. These observations suggest that, given more computational resources, Video-MTR has strong potential to scale to even larger frame budgets.
>
> **Table 1: Comparison between Qwen2.5-VL-7B and Video-MTR under different frame budgets.**
>
>
> | Model           | Frames | VideoMME(w/o sub.) | LongVideoBench Val | MLVU Test | LVBench| EgoSchema Test |
> |-----------------|:------:|:----------------------------:|:-------------------:|:---------:|:----------------:|:--------------:|
> | **Frames = 768**|        |                              |                     |           |                  |                |
> | Qwen2.5-VL-7B   | 768    | 65.2                         | 56.0                | 51.4      | 45.3             | 65.0           |
> | **Frames = 80** |        |                              |                     |           |                  |                |
> | Qwen2.5-VL-7B   | 80     | 59.5                         | 48.4                | 45.2      | 33.6             | 63.5           |
> | Video-MTR       | 80     | 62.7                         | 57.1                | 50.4      | 42.3             | 68.8           |
> | **Frames = 64** |        |                              |                     |           |                  |                |
> | Qwen2.5-VL-7B   | 64     | 58.4                         | 47.0                | 41.8      | 33.7             | 62.6           |
> | Video-MTR       | 64     | 59.0                         | 54.8                | 49.8      | 41.8             | 63.4           |
> | **Frames = 32** |        |                              |                     |           |                  |                |
> | Qwen2.5-VL-7B   | 32     | 53.6                         | 45.8                | 41.6      | 30.3             | 59.4           |
> | Video-MTR       | 32     | 62.2                         | 52.3                | 48.4      | 38.2             | 62.4           |
>
> [1] Longvideobench: A benchmark for long-context interleaved video-language understanding, Haoning Wu et.al 2024a.

---

> ### Author Response · Authors · 2025-11-20
>
> **Q2: Weird  QA accuracy in Figure 4**
> > It's weired that the QA accuracy in Figure 4 exceed 1. Some explanations should be provided.
>
> **A2**: We sincerely thank the reviewer for carefully checking Figure 4 and pointing out this error. This issue is caused by a mislabeling of the panel names: the left panel should show **QA accuracy**, while the right panel actually shows the **number of turns**, which explains why the values can be greater than 1. Our intention was to contrast QA accuracy and the number of turns to illustrate that, in the w/o goal-gated setting, the agent may increase its total reward simply by accumulating more turns, but with no corresponding gain in QA accuracy.
> We have corrected this error. We will also update the figure and its caption accordingly in the revised PDF version. We again apologize for the confusion caused by this mistake.

---

> ### Author Response · Authors · 2025-11-20
>
> **Q3: More information of compared baseline**
>
> >The information of compared baseline models is not given, especially their backbone models.
>
> **A3**: We thank the reviewer for the suggestion. We have therefore added Table 2 to give a more comprehensive comparison: in addition to the original parameter size and frame budget, we now also summarize, for each baseline, its **backbone LLM and post-training data scale**. We will integrate these full comparison from Table 2 in the Appendix of the revised version.
>
> Regarding the backbone comparison, Table 2 shows that our setting is fair across different implementation choices: Video-MTR shares the **exact same 7B Qwen2.5-VL-7B backbone with Video-R1 and uses the same 7B Qwen2 family** as LongVA and Video-XL, yet achieves superior performance while being trained on significantly less data (only an 8K long-video QA corpus). In contrast, many strong baselines rely on proprietary GPT-4/Gemini backbones or web-scale multimodal data.
> For the post-training data, to ensure a fair comparison, Table 2 reports only the data used in the post-training stage (instruction tuning or RL), rather than the full pre-training corpora. For this reason, we do not list the massive datasets used to build GPT-4/Gemini or the Qwen2.5-VL-7B backbone itself. Most counterparts rely on **hundreds of thousands to millions** of supervised multimodal pairs, whereas Video-MTR is post-trained in a single RL stage with **only 8K supervision-rich examples**, clearly highlights the strong data efficiency of our framework.
>
> **Table 2: Summary of compared baseline models, their backbones, frame budgets, and training data scale.**
>
> | Model                      | Params | Backbone (LLM)           | Post-training Data Scale      | Frames / fps      |
> |----------------------------|:------:|--------------------------|-------------------------------|-------------------|
> | GPT-4o (Hurst et al., 2024)        |   -    | GPT-4o (proprietary)     | -                             | 0.5 fps / 384     |
> | Gemini-1.5-Pro (Team et al., 2024) |   -    | Gemini (proprietary)     | -                             | 0.5 fps           |
> | DrVideo (GPT-4) (Ma et al., 2025)  |   -    | GPT-4 (proprietary)      | -                             | 0.2 / 0.5 fps     |
> | Qwen2.5-VL-7B† (Bai et al., 2025)  |  7B    | Qwen2.5-VL-7B            | -                             | 768               |
> | VideoLLaMA2 (Cheng et al., 2024)   | 8×7B   | Mixtral-8x7B-Instruct    | 1.35M                         | 8                 |
> | Video-CCAM (Fei et al., 2024)      |  9B    | Yi-1.5-9B-Chat           | 4.4M                          | 96                |
> | LongVA (Zhang et al., 2024)        |  7B    | Qwen2-7B-Instruct        | 760K                          | 128 / 256         |
> | Video-XL (Shu et al., 2025)        |  7B    | Qwen2-7B                 | 257K                          | 128 / 256         |
> | VideoAgent (Wang et al., 2024b)    |   -    | GPT-4 (proprietary)      | -                             | 87                |
> | VideoMemAgent (Fan et al., 2024)   |   -    | GPT-4 (proprietary)      | -                             | 72                |
> | Video-LLaVA (Lin et al., 2023)     |  7B    | Vicuna-7B-v1.5           | 765K                          | 8                 |
> | VideoChat2 (Li et al., 2024b)      |  7B    | Vicuna-7B-v0             | 2.0M                          | 16                |
> | LLaVA-OneVision (Li et al., 2024a) |  7B    | Qwen-2-7B                | 4.8M                          | 32                |
> | Video-R1 (Feng et al., 2025)       |  7B    | Qwen2.5-VL-7B            | 260K                          | 32 / 64           |
> | **Video-MTR (Ours)**               |  7B    | Qwen2.5-VL-7B            | **8K**                        | **32 / 64 / 80**  |

---

> ### Author Response · Authors · 2025-11-20
>
> **Q4: Efficiency analysis compared to baselines.**
> > While introducing multi-turn reasoning, the efficiency compared to baselines should be analyzed.
>
> **A4**: We thank the reviewer for asking about efficiency. Based on the updated Table 2, we analyze both inference and training efficiency.
> On the inference side, we use FLOPs as a quantitative proxy for computational cost, approximating $ \text{FLOPs} \propto (\text{model parameters} \times \text{number of tokens})$
>  , and for video-LLMs the number of tokens is roughly proportional to the **number of input frames**. Video-MTR uses the same 7B backbone as Qwen2.5-VL-7B but operates with a much smaller frame budget (32/64/80 frames), whereas some strong long-video baselines often rely on substantially larger inputs or models, e.g., 768-frame Qwen2.5-VL-7B†, 96-frame Video-CCAM (9B), 128/256-frame LongVA and Video-XL, or 8×7B VideoLLaMA2. Among 7B models with ≤96 input frames, Video-MTR achieves a favorable balance between accuracy and computational cost. Although Video-MTR introduces multi-turn reasoning, each turn only processes a small number of frames (32/16/8), so the **total number of processed frames per query remains well below** these large-frame settings. In practice, the average number of turns is modest (on the avgrage of 2.2 steps), so the overall FLOPs are comparable to or lower than those of single-turn long-video baselines with 96–768 frames.
> On the training side, Video-MTR is also highly efficient. Many baselines perform post-training on hundreds of thousands to millions of multimodal samples (e.g., 4.8M for LLaVA-OneVision, 4.4M for Video-CCAM, 2.0M for VideoChat2, 760K for LongVA, 260K for Video-R1), whereas our method uses only **8K** long-video QA episodes for RL post-training on the same 7B backbone. In practice, **1–2 epochs** of RL training are sufficient for the model to acquire effective multi-turn reasoning, making the training procedure very resource-friendly. Despite this drastic reduction in post-training data scale and training time, Video-MTR still outperforms these baselines on long-video benchmarks. This demonstrates that our multi-turn reasoning framework is not only competitive in performance, but also efficient in both inference and training.

---

> ### Author Response · Authors · 2025-11-20
>
> **Q5: Concerns about tricks of the training framwork**
> > The training framework is complicated with various tricks, which may be unstable in other scenarios.
>
> **A5**: We thank the reviewer for raising this concern. In fact, the core of our framework is simply the **gated bi-level reward system**, which is rooted in the standard practice of combining **process-level and outcome-level supervision in classical reinforcement learning**. Beyond this, we only introduce a small early-bootstrapping reward during the initial rollouts to encourage exploration, which is also a common RL technique. The policy is trained with the standard PPO algorithm using default hyperparameters. Overall, these choices make our training recipe conceptually simple and practically robust, rather than a collection of ad-hoc tricks that are fragile across scenarios.
>
> Regarding the gated bi-level reward, this is our main technical component. We design a temporally sensitive turn-level reward specifically for LVD, and using a gating mechanism to keep it aligned with the final QA objective. The reward configuration itself is **straightforward and follows prior work such as DeepSeek** [2]: we set the QA accuracy reward to 1.0 and the format reward to 0.1; the additional turn-level reward introduced by our method is fixed to 0.5 (half of the QA reward), with no per-dataset tuning.
>
> For the early-bootstrapping reward, we explicitly tested different tool-use bonus values (0.1 / 0.2 / 0.5) and found that they primarily affect the convergence speed of learning to call tools, while having negligible impact on the final QA accuracy. This confirms that this reward term is not overly sensitive to specific threshold choices.
>
> [2] Deepseek-r1: Incentivizing reasoning capability in llms via reinforcement learning，Daya Guo et al., 2025
>
> [3] Pixel Reasoner: Incentivizing Pixel-Space Reasoning with Curiosity-Driven Reinforcement Learning，Haozhe Wang et al., 2025
>
> [4]DeepEyes: Incentivizing “Thinking with Images” via Reinforcement Learning,Ziwei Zheng et al., 2025

---

> ### Author Response · Authors · 2025-11-24
>
> Dear Reviewer R4Fn,
>
>  We have submitted a revised version of the manuscript along with detailed responses to your comments. Could you please have a look at our response? We would greatly appreciate any further feedback and are looking forward to your reply.
>
> Authors of Video-MTR

---

> > ### Comment · Reviewer_R4Fn · 2025-11-26
> >
> > Thanks for your response. I have some minor problems. What's the VRAM cost in training with different numbers of frames? Moreover, what's your fine-tuning setting (e.g., LoRA parameters)?

---

> > > ### Author Response · Authors · 2025-11-26
> > >
> > > Dear Reviewer R4Fn，
> > >
> > > Thank you for your question and continued engagement. Below we provide per-GPU VRAM usage for different frame budgets and outline our fine-tuning configuration.
> > >
> > > **1、VRAM Analysis (training)**
> > >
> > > Our experiments were conducted on a single server equipped with **8 × NVIDIA A800-80G** , utilizing the VerL training framework. The numbers below report **per-GPU** settings and observed **VRAM usage (avg / peak)** for different input frame budgets. The table highlights two critical memory observations: (i) Average Cost Escalation: Keeping a constant per-GPU batch size (Frames 32 to 64) increases the average vRAM cost.(i) **Peak Pressure from Length**: To prevent OOM at the longest sequence (80 Frames), the batch size was halved, yet peak vRAM utilization still rose, underscoring the transient memory pressure of extended inputs.
> > >
> > > **Table 3. Training Details and vRAM Consumption with Varying Frame Budgets (Per GPU).**
> > > | Frame Budget | Per-GPU Batch Size | Average vRAM Cost (GB) | Peak vRAM Usage (GB) |
> > > | :----------: | :----------------: | :--------------------: | :------------------: |
> > > | 32 | 4 | 33 | 58 |
> > > | 64 | 4 | 38 | 67 |
> > > | 80 | 2 | 32 | 75 |
> > >
> > > **2、 Fine-tuning setting**
> > >
> > > We employ **full parameter fine-tuning** rather than parameter-efficient methods. To make full-parameter training feasible, we use FSDP, gradient checkpointing, and a memory-aware rollout setup. Key configurations are as follows:
> > >
> > > **Optimization**
> > >
> > > - Actor LR = 1e-6, Critic LR = 1e-5.
> > > - PPO mini-batch size = 8.
> > > - KL regularization enabled with KL-coef = 1e-3 (MSE-based KL).
> > > - Gradient checkpointing enabled for both actor and critic.
> > >
> > > **Model parallelism & memory**
> > > - Total Batch size：32/16
> > > - vLLM-based rollout for efficiency, with chunked prefill and memory utilization capped at 0.3/0.25.
> > >
> > > **Training configuration**
> > > - Total training steps = 600.
> > > - The rollout model uses top-p = 0.95 and temperature = 0.7.
> > > - Full-sequence training with remove-padding optimization and max model length controlled by MAX_MODEL_LEN=8192.

---

> > > > ### Comment · Reviewer_R4Fn · 2025-11-28
> > > >
> > > > Thanks for your reply. Though some of my concern are not fully addressed, I'd like to raise my rating.

---

> > > > > ### Comment · Reviewer_R4Fn · 2025-11-28
> > > > >
> > > > > I cannot edit my review now. Is there a bug???

---

> > > > > ### Author Response · Authors · 2025-11-28
> > > > >
> > > > > Dear Reviewer R4Fn，
> > > > >
> > > > > Thank you very much for your positive assessment and for your willingness to revise the rating, and we truly appreciate it.
> > > > > It appears the system is temporarily restricting modifications. You might try checking again a bit later.

---

### Official Review · Reviewer_StqD · 2025-11-01

**Soundness:** 3
**Presentation:** 3
**Contribution:** 3
**Rating:** 6
**Confidence:** 3

**Summary:**

The paper proposes Video-MTR, a reinforcement-learning based framework for multi-turn reasoning in long video understanding. The main idea is to start with uniform frame sampling, and then use an MLLM to iteratively decide whether to retrieve additional frames or answer the question. The policy is optimized using PPO with a gated bi-level reward that combines final-answer correctness and turn-level frame-query relevance. Experiments on VideoMME, MLVU, and EgoSchema show improvements over existing open-source baselines on long videos while using relatively few training examples.

**Strengths:**

- The paper studies the under-explored idea of combining reinforcement learning with long-video understanding, and the proposed method is novel. The empirical results are quite strong and Video-MTR beats a number of competitive baselines.
- The paper is mostly clearly written and easy to read. The proposed method is well-motivated.
- The paper conducts a number of ablation experiments, and in addition to QA accuracy it also measures latency as an additional metric in some experiments in the appendix.

**Weaknesses:**

- The design of the reward function seems to be a bit ad-hoc. It would be useful to know how sensitive the method is to hyper‐parameters (thresholds and bonus amounts in each stage).
- The paper is lacking some error analysis and discussion on failure modes, e.g. when wrong segments are retrieved.

There are a few occasions in the paper where there might be ambiguities or inaccurate claims, and I would hope that they can be clarified in the paper:
- Around line 78 the authors claim that "this is the first attempt to incorporate multi-turn reasoning in the context of long video understanding." This claim is a bit inaccurate since one can argue that the large body of existing works in "agentic" video models (such as "VCA: Video Curious Agent for Long Video Understanding", Arxiv 2412.10471) are also doing multi-turn exploration and refinement of the selected video frames within a long video.
- Around line 335 the authors claim that "Most open-source long-video methods operate with ≤ 128 frames." It used to be the case but today more and more models support longer context. For example the Qwen2.5-VL-7B model that the paper refers to officially supports a context length of 131072 and 768 frames when processing videos.

**Questions:**

I would like the authors to discuss the following concerns I have on the methodology and the experiments. They are not necessarily weaknesses of the paper, but rather questions I would like to gather more information on from the authors:
- The dataset the authors curated has a size of only 8K, which might be a bit too small for long videos. Do the author agree that it is a valid concern that policy may rely on heuristics tied to those training datasets rather than truly understand how to retrieve, and might not work well on new video domains?
- For ultra long videos for questions that require complex reasoning and fine-grained understanding at multiple locations of the long video, retrieving only 32/64 frames and only using 3 turns might not be sufficient. Do you have a sense on why increasing these numbers did not lead to uniform increase in performance in your experiments?

---

> ### Author Response · Authors · 2025-11-20
>
> Dear Reviewer StqD,
> We sincerely thank the reviewer for the positive assessment, and for the clear and constructive comments. We first respond to the specific items listed under Questions. We then address the issues raised under Weaknesses one by one, aiming to clarify any ambiguities and provide additional details.
>
> **Q1： Data scale & generalization**
> > The dataset the authors curated has a size of only 8K, which might be a bit too small for long videos. Do the author agree that it is a valid concern that policy may rely on heuristics tied to those training datasets rather than truly understand how to retrieve, and might not work well on new video domains?
>
> **A1** : We thank the reviewer for raising this concern. Our strictly **out-of-domain evaluation** already strongly suggests that Video-MTR learns a **generalizable policy across different video domains**. We train only on NExT-QA and QVHighlights, and evaluate on strictly disjoint benchmarks (VideoMME, EgoSchema, MLVU, LVBench). The consistent performance gains over the Qwen2.5-VL-7B backbone across these unseen distributions indicate that our policy transfers well, rather than memorizing dataset-specific shortcuts.
>
> Crucially, it is the RL-based training that enables this transferable reasoning behavior. Recent studies on post-training paradigms, such as “SFT Memorizes, RL Generalizes” [1] and “Good Actions Succeed, Bad Actions Generalize” [2], show that SFT often memorizes the training distribution and struggles in out-of-distribution scenarios, whereas outcome-based RL tends to learn more robust, generalizable behaviors. In line with these findings, our goal-gated reward, encouraging the model to acquire transferable reasoning rules through exploration rather than memorizing dataset-specific patterns.
> In our setting, this is further validated by the comparison with Video-R1: it shares the same Qwen2.5-VL-7B base model, but relies primarily on large-scale supervised fine-tuning, with only a brief RL phase as a secondary refinement stage. On EgoSchema, a distinct dataset composed entirely of egocentric viewpoints—Video-R1’s performance drops to 48.8%, even below the backbone’s 59.4%, suggesting overfitting to its supervised training distribution. In contrast, our purely RL-based multi-turn scheme reaches 62.4% accuracy on EgoSchema, second only to GPT-4o and Gemini-1.5-Pro. Notably, this strong performance is achieved **without any additional egocentric data** during training, indicating that RL helps the model move beyond dataset-specific heuristics and learn more generalizable retrieval and reasoning behaviors.
>
> [1] SFT Memorizes, RL Generalizes: A Comparative Study of Foundation Model Post-training. Tianzhe Chu, Yuexiang Zhai et al., ICLR 2025
>
> [2] Good Actions Succeed, Bad Actions Generalize: A Case Study on Why RL Generalizes Better, Meng Song , arxiv 2025

---

> ### Author Response · Authors · 2025-11-20
>
> **Q2： Non-uniform gains when increasing frames/turns**
>
> **A2** : We thank the reviewer for raising this question. Upon analyzing this phenomenon, we beleive that the non-uniform gains in our experiments are not a fundamental limitation of the multi-turn reasoning framework itself, but rather a consequence of the **mismatch between available training data and evaluation benchmarks**.
>
> In LVD,  a clear gap exists: recent evaluation benchmarks are explicitly designed to push the limits of MLLMs with multi-event, minute-to-hour videos, while existing training datasets still contain relatively few ultra-long and truly challenging videos. This disparity is summarized in **Table 1 (evaluation benchmarks) and Table 2 (commonly used training datasets)**.
> In our setup, we follow the common practice in the community and do not train on the dev/test-style splits of these evaluation benchmarks (e.g.,  MLVU dev，LongVideoBench val）, ensuring that each benchmark remains a new domain for fair generalization comparison. As a result, the policy sees limited supervision on truly challenging cases during training. Our chosen configuration (32/64 frames, 3 turns) is therefore a practical balance given the current training corpus.
> Importantly, despite the significant gap in video length between training data and test benchmarks, the model still learns a useful multi-turn reasoning pattern:  yield clear improvements over the single-turn Qwen2.5-VL-7B backbone. We believe that as larger and more diverse long-video training datasets  become available, our multi-turn retrieval and reasoning framework can further benefit from scaling up the frame and turn budgets and is likely to exhibit more consistent improvements on ultra-long video scenarios.
>
>
> **Table 1: Long-video evaluation benchmarks (mainly used for testing, rarely for training).**
>
> | Benchmark      | #Videos | #QAs (split)                                      | Avg. length (s) |
> |----------------|:-------:|---------------------------------------------------|:---------------:|
> | VideoMME       |   900   | 2,700                                             |      1024       |
> | MLVU           |  1,730  | 3,102 (dev: 2,573; test: 509)                     |       930       |
> | LongVideoBench |  3,763  | 6,678 (val: 752; test: 3,011; others: 2,915)      |       473
> | LVBench        |   103   | 1,549                                             |      4101       |
>
>
> **Table 2: Commonly used training datasets for long-video models.**
> - Our 8K long-video QA training examples for **Video-MTR** are constructed from subsets of **NExT-QA** and **QVHighlights**.
>
> | Dataset            | #Videos     | #QAs / Instructions (split)                                                     | Avg. length (s) |
> |--------------------|------------:|---------------------------------------------------------------------------------|:---------------:|
> | LLaVA-Video-178K   | 178K videos | 960K open-ended QAs, 196K MC QAs                                               |     0–180       |
> | NExT-QA            | 5K videos   | 52K manually annotated QAs (train: 3,880; val: 570; test: 1,000)              |       44        |
> | QVHighlights       | 10K videos  | 10,310 queries associated with 18,367 moments                                  |      150        |

---

> ### Author Response · Authors · 2025-11-20
>
> Next, we address the remaining questions and concerns raised by the reviewer in the "Weaknesses" section:
>
> **Weekness-Q1：Reward design & hyper-parameter sensitivity**
> > The design of the reward function seems to be a bit ad-hoc. It would be useful to know how sensitive the method is to hyper‐parameters (thresholds and bonus amounts in each stage).
>
> **A1**：We thank the reviewer for raising this concern. Our core reward function design is not ad-hoc, but is rooted in the common practice of integrating **process supervision and outcome supervision** in classical Reinforcement Learning problems. We adapted this by designing a temporally-sensitive turn-level reward specifically for LVD, using a gated mechanism to ensure the primacy of the final result reward. Our reward configuration is also **straightforward and follows prior work** such as DeepSeek [3]. We set the final QA accuracy reward to 1.0 and the format reward to 0.1. The additional turn-level reward introduced by our method is empirically fixed to 0.5, deliberately set at half the QA reward and above the format reward. Importantly, our analysis demonstrated that the method is insensitive to the specific magnitude of this process reward. We tested different values (e.g., 0.3, 0.5, and 0.7) and found no significant impact on the final performance. Furthermore, we use this same setting across all datasets and experiments without any per-dataset tuning.
> The only additional bonus we include is for tool usage.  This kind of early exploration encouragement (via small bonus rewards) is also a common technique in reinforcement learning. We explicitly tested different tool-use bonus values (0.1 / 0.2 / 0.5) and found that they primarily affect the model's convergence speed in learning to call tools, yet have negligible impact on the final QA accuracy.
> Taken together, these observations indicate that the overall performance and conclusions are stable under a reasonable range of reward coefficients. This confirms that our reward design, while simple, is reasonably robust rather than ad-hoc, and is not overly sensitive to specific threshold or bonus choices.
>
> [3] Deepseek-r1: Incentivizing reasoning capability in llms via reinforcement learning，Daya Guo et al., 2025

---

> ### Author Response · Authors · 2025-11-20
>
> **Weekness-Q2：Failure Modes Discussions**
> > The paper is lacking some error analysis and discussion on failure modes, e.g. when wrong segments are retrieved.
>
> **A2**：Thanks for the comment. Due to space limitations, we could not include a detailed discussion of failure modes in the main text. Instead, in **Appendix C (Case Studies)** we present additional examples from three evaluation benchmarks to illustrate Video-MTR’s multi-turn reasoning process, including both successful and failed cases. We explicitly analyze failure cases to diagnose error sources and outline potential remedies. These cases reveal several structural limitations of the current Video-MTR in complex scenarios. Based on these observations, we discuss the need for deeper temporal search policies and hierarchical zoom-in vision modules to more reliably handle multi-event reasoning and fine-grained perception in future work.
>
>
> **Weekness-Q3: Clarification of the “first” multi-turn reasoning claim and relation to agentic video models**
>
> **A3**: We thank the reviewer for pointing out this ambiguity.  Our intended scope was specifically that Video-MTR is **the first reinforced multi-turn reasoning framework for Long Video Understanding (LVD)** that employs end-to-end policy training, rather than the first work to use multi-turn exploration generally.  We will update the wording in the revised version to: "we introduce Video-MTR, to our knowledge the first RL-based multi-turn reasoning framework for long video understanding with end-to-end policy training." We hope this clarification clearly positions our contribution relative to existing agentic video models.
> As discussed in the Introduction (Lines 48–50), we acknowledge existing agentic models (e.g., VideoAgent, VideoTree), which incorporate multi-step exploration. However, these systems are typically **training-free or heuristic frameworks** that rely on heterogeneous external components and sub-optimal tool usage strategies. For instance, models like VCA build exploration strategies and reward models but do not perform end-to-end policy optimization of the underlying model parameters.
> In sharp contrast, Video-MTR is a unified framework where the entire multi-turn reasoning policy is optimized end-to-end using Policy Gradient RL based on a long-horizon reward. This key innovation, the **RL-based, end-to-end policy optimization** for multi-turn reasoning, is precisely our primary contribution to the LVD literature.
>
> **Weekness-Q4：Clarification on “≤128 frames” for open-source long-video methods**
>
> **A4**: We apologize for the imprecise wording of our claim. Our statement was intended to refer specifically to open-source long video understanding (LVD) **post-training methods** developed in the academic community, rather than to large-scale industrial foundation models. As shown in our Table 4, the majority of these LVD post-training methods indeed operate with input frame budgets of ≤128 frames, with only a few exceptions that employ specialized input-sequence length optimization (e.g., Video-XL, LongVA, which are still bounded at 256 frames). By contrast, the powerful general-purpose VLM foundation models (such as Qwen2.5-VL-7B, GPT-4o, and Gemini) rely on massive, often unprecedented, computational resources during pre-training, which inherently enables them to support extremely long context lengths.

---

> ### Comment · Reviewer_StqD · 2025-11-20
>
> I appreciate the authors' comments for the additional results and clarifications. Those responses addressed my concerns. Considering the novelty of the method and the overall quality of the experiments, I increased my score to from 6 to 8.

---

> > ### Author Response · Authors · 2025-11-21
> >
> > Thank you very much for your thoughtful follow-up. We truly appreciate your updated score and are glad that the additional results and clarifications addressed your concerns.

---

### Author Response · Authors · 2025-11-24
**Key Changes in the Revised Version**

Dear Reviewers,

We thank the reviewers for their time and careful evaluation of our manuscript. Their insightful comments and constructive feedback motivated us to run additional experiments and provide further clarifications, which have substantially improved the quality and rigor of the paper.
Notably, to directly address the reviewer’s request to investigate whether Video-MTR can outperform Qwen2.5-VL with more frames, we extended our experiments to larger frame budgets. The new results show that it brings further performance gains, and **Video-MTR (80 frames)** even outperforms it on **EgoSchema (+3.8%) and LongVideoBench (+1.1%)**.

In addition to the responses, we have submitted a revised manuscript, along with an appendix that lists all changes, with revisions highlighted in magenta for clarity. Due to space constraints, only selected new experiments are included in the main paper, while the remaining supplementary results are provided in the appendix：
*  **Extended input frame budgets** . We extended our experiments to a larger frame budget of 80 frames, which brings further performance gains. The updated comparisons are reported in Tables 1 and 3.
*  **Extended Benchmark Comparison** . We expanded the evaluation by adding LongVideoBench, a commonly used long-video understanding benchmark that is also reported in the Qwen2.5-VL technical report. The new comparison results are included in Table 1.
*  **Extended Comparisons of compared baseline** .  To give a more comprehensive comparison, we extended the baseline comparison metrics. In addition to the original parameter size and frame budget, we now also summarize, for each baseline, its backbone LLM and post-training data scale. We integrate these full comparisons in Appendix D .
*  **Correction of reward-hacking figure** .  We fixed the mislabeling of panel names in the reward-hacking example (Figure 4): the left panel now correctly shows QA accuracy , while the right panel shows the number of intermediate turns . We have updated both the figure and its caption accordingly in the revised manuscript.

We believe that the revisions made to the manuscript, along with the detailed rebuttals, have significantly improved its quality and addressed the concerns raised.

---

### Author Response · Authors · 2025-12-01
**Summary of Rebuttal Progress and Reviewer Discussions**

Dear AC,

In this section, we briefly reiterate the main contributions of our work and summarize the key issues addressed during the rebuttal, along with the outcome of our discussions with the reviewers.

**Main Contributions**

We propose Video-MTR, which to the best of our knowledge, is the **first reinforced multi-turn reasoning framework for Long Video Understanding (LVD)**  featuring end-to-end policy training. Extensive experiments on five mainstream LVD benchmarks (VideoMME, MLVU, LongVideoBench, LVBench, and EgoSchema) demonstrate that Video-MTR outperforms existing methods in both accuracy and efficiency under comparable settings.

**Key Additions During Rebuttal**

- Enhanced Performance: We conducted additional experiments that further boosted Video-MTR’s performance. The new results show that Video-MTR (using only 80 frames) achieves performance comparable to Qwen2.5-VL-7B (using 768 frames) across five benchmarks, and even outperforms it on EgoSchema and LongVideoBench. This strongly validates the superiority of our multi-turn reasoning framework.
- Comprehensive Efficiency Analysis: We provided a extended comparison against all baselines, demonstrating that our framework is highly efficient in both inference and training.
- Mechanism Clarification: We elaborated on the training framework mechanism, clarifying that our training recipe is conceptually simple and practically robust.

All updates and new findings described below have been incorporated into the revised version, which has substantially improved the quality and rigor of the paper.

**Outcome of Reviewer Discussion**

After the discussions, **all reviewers expressed that most of their concerns had been addressed** and indicated their willingness to increase their scores. Consequently, the paper has received **consistently positive ratings** (8, 6, and 6).
We regret that the early closure of the rebuttal phase and the subsequent revert to official reviews. Nevertheless, we are deeply grateful for the reviewers’ additional efforts during the rebuttal period and for their positive assessment of our work.


Sincerely，

Authors of Video-MTR

---

### Meta-Review · Area_Chair_J7D3 · 2026-01-07

**Summary:**

The paper proposes a reinforced multi-turn reasoning framework for long-video understanding and presents extensive experimental results across several benchmarks.

**Reviewer Concerns:**

The reviews acknowledge the novelty and empirical strength of the approach, but raise substantive concerns regarding the clarity and strength of the novelty claims relative to prior agentic and modular methods, the reliance on additional temporal supervision for turn-level rewards, and remaining questions about robustness, scalability, and generality beyond the evaluated settings. While the authors provided a detailed rebuttal and additional experiments, not all of the reviewers’ conceptual and methodological concerns appear to be fully resolved.

**Reviewer Scores:**

Based on the remaining issues highlighted in the reviews and the fact that the rebuttal does not fully address them, the recommendation is to reject the paper at this time.

---

### Decision · Program_Chairs · 2026-01-26

Reject